# External Validation of a Convolutional Neural Network for IDH Mutation Prediction

**DOI:** 10.3390/medicina58040526

**Published:** 2022-04-09

**Authors:** Iona Hrapșa, Ioan Alexandru Florian, Sergiu Șușman, Marius Farcaș, Lehel Beni, Ioan Stefan Florian

**Affiliations:** 1Department of Medical Genetics, Iuliu Hațieganu University of Medicine and Pharmacy, 8 Victor Babes Street, 400012 Cluj-Napoca, Romania; i.hrapsa@gmail.com; 2Department of Neurosurgery, Iuliu Hațieganu University of Medicine and Pharmacy, 8 Victor Babes Street, 400012 Cluj-Napoca, Romania; stefanfloriannch@gmail.com; 3Department of Neurosurgery, Emergency County Hospital, 3-5 Clinicilor Street, 400006 Cluj-Napoca, Romania; lehelbeni@yahoo.com; 4Department of Morphological Sciences-Histology, Iuliu Hațieganu University of Medicine and Pharmacy, 8 Victor Babes Street, 400012 Cluj-Napoca, Romania; serman_s@yahoo.com; 5Department of Pathology, IMOGEN Research Center, Louis Pasteur Street, 400349 Cluj-Napoca, Romania; 6Department of Genetics, IMOGEN Research Center, Louis Pasteur Street, 400349 Cluj-Napoca, Romania

**Keywords:** radiomics, radiogenomics, artificial intelligence, IDH, glioma, convolutional neural network, prediction

## Abstract

*Background and Objectives*: The IDH (isocitrate dehydrogenase) status represents one of the main prognosis factors for gliomas. However, determining it requires invasive procedures and specialized surgical skills. Medical imaging such as MRI is essential in glioma diagnosis and management. Lately, fields such as Radiomics and Radiogenomics emerged as pertinent prediction tools for extracting molecular information out of medical images. These fields are based on Artificial Intelligence algorithms that require external validation in order to evaluate their general performance. The aim of this study was to provide an external validation for the algorithm formulated by Yoon Choi et al. of IDH status prediction using preoperative common MRI sequences and patient age. *Material and Methods*: We applied Choi’s IDH status prediction algorithm on T1c, T2 and FLAIR preoperative MRI images of gliomas (grades WHO II-IV) of 21 operated adult patients from the Neurosurgery clinic of the Cluj County Emergency Clinical Hospital (CCECH), Cluj-Napoca Romania. We created a script to automate the testing process with DICOM format MRI sequences as input and IDH predicted status as output. *Results*: In terms of patient characteristics, the mean age was 48.6 ± 15.6; 57% were female and 43% male; 43% were IDH positive and 57% IDH negative. The proportions of WHO grades were 24%, 14% and 62% for II, III and IV, respectively. The validation test achieved a relative accuracy of 76% with 95% CI of (53%, 92%) and an Area Under the Curve (AUC) through DeLong et al. method of 0.74 with 95% CI of (0.53, 0.91) and a *p* of 0.021. Sensitivity and Specificity were 0.78 with 95% CI of (0.45, 0.96) and 0.75 with 95% CI of (0.47, 0.91), respectively. *Conclusions*: Although our results match the external test the author made on The Cancer Imaging Archive (TCIA) online dataset, performance of the algorithm on external data is still not high enough for clinical application. Radiogenomic approaches remain a high interest research field that may provide a rapid and accurate diagnosis and prognosis of patients with intracranial glioma.

## 1. Introduction

Undeniably, medical imaging has revolutionized noninvasive medical diagnosis across all fields of medicine, neurosurgery included. The spectacular progress in efficiency and processing power of personal computers has enabled the transition from visual to computer aided imaging interpretation. Owing to the increasing image processing capabilities, computer algorithms are able to yield diagnostic information with accuracies comparable to radiology specialists [1]. Furthermore, research is focusing on more integrated ways that may predict and help the clinician with valuable diagnostic or prognostic information in a faster and cheaper manner with the help of Artificial Intelligence (AI) algorithms. This field of research is known as Radiomics, a radiological subject area wherein images are converted into minable data and then into practical information in a computer [2].

In neuro-oncology, imaging is a necessity for accurate and effective treatment. However, a precise and definitive diagnosis can only be established after biopsy or tumoral resection, followed by histopathological interpretation and molecular profiling. Although biopsy is a minimally invasive procedure, there are nonetheless several risks concerning sampling errors (due to tumor heterogeneity) or surgery-related complications [3]. In 2016, the World Health Organization (WHO) has finally introduced molecular parameters for the classification of central nervous system (CNS) tumors, emphasizing the need and the clinical importance of genetic profiling [4].

Isocitrate dehydrogenase (IDH) is a Krebs cycle enzyme that plays a crucial role in cellular homeostasis. Mutations in its respective genes (IDH1, IDH2) have been associated with oncogenesis and a particular cancer metabolism that poses a potential therapeutic target. Concurrently, the presence of IDH mutations offer valuable prognostic and diagnostic information [5].

Radiogenomics is a cross field between two ‘omics’ or data-oriented fields: Radiomics and Genomics. Its main goal is to integrate genomic data obtained from radiomic features. Concurrently, the molecular profile of a tumor is viewed as a whole, with radiogenomics aiming to reduce sampling errors, anticipate biopsy results and prognosis, and spare medical resources. Using radiogenomic approaches on brain tumor MRI images, IDH1 mutations have been accurately predicted in several articles making this a veritable alternative to biopsy [6,7,8,9].

This article focuses on a recent study: “Fully automated hybrid approach to predict the IDH mutation status of gliomas via deep learning and radiomics” by Yoon Choi et al. [6]. We have chosen to follow this article since it has a promising performance, the code is open source, it is very well documented, and can be executed on a personal computer. However, as with any Machine Learning (ML) algorithm, it requires an external validation in order to appreciate its actual performance. The aim of this study is to validate the efficiency and accuracy of the algorithm furthered by Choi et al. on local imaging data from the Neurosurgery clinic database of the Cluj County Emergency Clinical Hospital (CCECH), Romania.

## 2. Materials and Methods

### 2.1. Algorithm Description

The aim of the study by Choi and associates was to develop a radiogenomic model to predict IDH mutation status based on magnetic resonance imaging (MRI) sequences, specifically T1 contrast-enhanced (T1c), T2, and fluid-attenuated inversion recovery (FLAIR) in gliomas of grades WHO II-IV [6]. Succinctly, the algorithm is comprised of 3 components: an image processing part (where the 3 sequences are aligned, rescaled, and intensity normalized) and two Convolutional Neural Network (CNN) models, which are a type of specialized image processing AI algorithms. Model 1 was tasked with performing the segmentation of the whole tumor volume, while Model 2 was trained to predict the IDH mutation status based on the segmented tumor, its radiomics features (of shape and loci), and patient age. Choi et al. trained the two models on 727 patient datasets from a local Korean hospital, then tested them on a different local hospital dataset and on an online dataset from The Cancer Imaging Archive (TCIA) [10]. The code and a manually run pipeline were then published on GitHub [11] at https://github.com/yoonchoi-neuro/automated_hybrid_IDH, accessed on 23 February 2021.

### 2.2. Sampling-Inclusion and Exclusion Criteria

We considered 114 adult patients with newly diagnosed gliomas grades WHO II-IV operated between 2018–2020 from the patient database of Neurosurgery clinic of CCECH. Interventions were performed by the senior neurosurgeon of the department (I.S.F.). The following including criteria were applied: (1) confirmed histopathological diagnosis (2) Preoperative T1c, T2, FLAIR MRI images (3) Molecularly (MLPA-Multiplex PCR) or immunohistochemically (IHC) determined IDH status (4) Age above 18. The exclusion criteria were as follows: (1) History of brain surgery (*n* = 7) (2) Absence of either T1c, T2, or FLAIR imaging modalities (*n* = 45) (3) Unknown or inconclusive IDH mutation status (*n* = 41). Eventually, 21 patients remained eligible for the study (Figure 1).

### 2.3. Data Collection and Database Creation

Entire imaging, clinical and biological data was collected retrospectively. The 3 MRI modalities were obtained on 1.5T equipment (62%-GE MEDICAL SYSTEMS, 29%-SIEMENS and 9%-TOSHIBA). A complete list of manufacturers and models can be viewed in the Appendix A. Slice thickness were 1–2 mm (71%) and 2–5 mm (29%) for T1c; 4–5 mm (81%) and 1–3 mm (19%) for T2; 4–5 mm (90%) and <4 mm (10%) for FLAIR. IDH status was determined through IHC (*n* = 16) and MLPA (*n* = 5). Imaging data files in DICOM (Digital Imaging and Communications in Medicine) format were collected from the PACS (Picture Archiving and Communication System) archive of the clinic, whilst clinical (diagnosis, age, sex) and molecular data (IDH mutation status) were collected from the AtlasMed integrated hospital database. Subsequently, imaging data was linked to patient clinic-biological data and an anonymized database was created.

#### 2.3.1. Software and Hardware Requirements

In terms of hardware, we used an ASUS ROG GL552JX-DM019D gaming laptop with Intel^®^ Core™ i7-4720HQ, 2.60 GHz, 8 GB RAM and Nvidia GeForce GTX 950M 4 GB GPU running a Linux based distribution (Ubuntu 20.04 LTS).

In terms of software, the code requires a set of libraries and tools: Python3-the base code interpreter; *FMRIB’s Linear Image Registration Tool* (*FLIRT)* [12] from *FMRIB Software Library* (*FSL)* [13]-for the image aligning part; *Advanced Normalization Tools (ANTs)* [14] for the intensity normalization part; Brain Extraction Tool *(BET)* [15] from *FSL*-for skull removing and brain extraction; *Nipype* [16]-a Python package to interface with the *FLIRT*, *BET* and *ANTs* tools; PyRadiomics [17]-a Python package to extract radiomic features; *PyTorch* [18]-a Python package used to run the two CNN trained models. The algorithm also requires a CUDA (Compute Unified Device Architecture) [19] compatible GPU from Nvidia to take advantage of the parallel computation.

#### 2.3.2. Image Preprocessing, Image Processing, Model 1, Model 2

The 3 imaging modalities in DICOM format were converted into NIfTI (Neuroimaging Informatics Technology Initiative) format which is a 3D rendering of the image. Machine Learning algorithms can work much more efficiently with that in contrast to 2D slices of DICOM format [20]. Conversion was achieved using the *dcm2niix* tool [21]. During the conversion process metadata becomes automatically discarded and images become anonymized. Next, comes the skull stripping part with the *BET* tool that finally outputs the 3 files ready to feed Choi’s algorithm (Figure 2).

The 3 previously prepared files are then processed through the FLIRT and ANTs tools and the tumor becomes segmented using the Model 1. Then, based on this segmentation, radiomic features are extracted (using *PyRadiomics*) and together with T1c and T2 images and patient age are fed into Model 2, which then predicts the IDH status.

To automate this process for every patient, we created a Python script that takes DICOM files as input, does the NIfTI conversion, applies BET, runs Choi’s algorithm, takes up the age from the previously created database and outputs the predicted IDH status into a csv (comma separated file) file next to patient’s ID (Figure 3). The script can be accessed at: https://github.com/ihrapsa/IDHpredict, accessed on 23 February 2021.

### 2.4. Statistics Software and Tests

The obtained predictions and the IDH status obtained from molecular and IHC tests were compared into a 2 × 2 contingency table. Sensibility, Specificity, Positive Predictive Value (PPV), Negative Predictive Value (NPV) and Accuracy were calculated. At the same time, based on the absolute prediction values, Receiver Operating Characteristic (ROC) curve was drawn, and the Area Under the Curve (AUC) was determined together with its *p* value using DeLong et al. method. All the tests and graphs were carried out using version 9 of Prism from GraphPad.

## 3. Results

Study sample characteristics as well as a comparison with Severence training dataset can be observed in Table 1.

Per patient run took around 50 min and the script finished testing the entire sample in roughly 19 h. The created contingency table can be observed in Table 2.

The relative accuracy of the prediction was 76% with 95% CI of (53%, 92%), Sensitivity was 0.78 with 95% CI of (0.45, 0.96), Specificity was 0.75 with 95% CI of (0.47, 0.91). PPV was 0,7 with 95% CI of (0.4, 0.89) while NPV was 0.82 with 95% CI of (0.52, 0.97). The ROC curve can be observed in Figure 4. The calculated AUC through DeLong et al. method was 0.74 with 95% CI of (0.53, 0.91) and a *p* of 0.021.

## 4. Discussion

In this study, we used a trained AI algorithm to determine the IDH status of 21 adult patients operated for gliomas WHO grades II-IV within the CCECH-department of neurosurgery, based on their MRI imaging and age. For external validation, the authors of the code (Choi et al.) tested the algorithm on both a case series from a different hospital (Seoul National University Hospital–SNUH), as well as cases from the online archive of TCIA [6]. We compared our results with those obtained from the abovementioned datasets (TCIA, SNUH), as our validation is also an external one.

The diagnostic performance of the algorithm on our cases is statistically significant (*p* < 0.05) with an AUC of 0.74 and Accuracy of 76%. Our values are closer to those obtained by Choi et al. in the TCIA dataset (AUC = 0.86, Accuracy = 78.8%) than the ones in the SNUH dataset (AUC = 0.94, Accuracy = 87.9%).

The differences observed could have stemmed from multiple causes: differences between the quality and acquisition resolution of the MRI images (100% 1.5 T in our study vs. 85% 3 T in SNUH and 44.3% 3 T in TCIA), widely different machines and imaging protocols, different patient characteristics, as well as a relatively small number of only 21 cases compared to 107 in SNUH and 203 in TCIA.

Although surgical resection is the golden standard first-line treatment of gliomas regardless of the IDH status, knowing the IDH status heightens the value of the prognosis, which can also dictate the subsequent therapeutic approach. It is a known fact that the presence of the mutation is associated with a better prognosis, regardless of the WHO grade [22]. Recently, Patel et al., while comparing the impact of resection on survival based on the molecular status of low-grade gliomas, noticed a more favorable response to total resection in IDH+ astrocitomas than in astrocitomas with a different genetic profile [23]. Furthermore, according to the update on the WHO classification of brain tumors from 2016, the absence of the IDH mutation excludes oligodendrogliomas, making it useful in the classification of these neoplasms [24].

There are multiple studies on similar topics, some of which have demonstrated remarkable performance. Notably, Bangalore Yoganada et al., after utilizing a 3D CNN for the classification, managed to obtain excellent results using only volumes extracted from T2 sections (AUC = 0.98 ± 0.01) [8]. An interesting aspect of their study is the segmentation of the tumoral volume based on the IDH status, knowing that within the structure of a tumor both IDH+ regions and IDH− regions may coexist. Calabrese et al. trained a model for the segmentation of glioblastoma that, based on the radiomic features it extracted, managed to detect the IDH, ATRX, and CDKN2 mutations, as well as aneuploidies of chromosomes 7/10, with AUCs ranging from 0.85 to 0.97 on their internal validation dataset. [25]. Lu et al., using a ML algorithm (Support Vector Machine), managed to differentiate between Glioblastoma and Low-Grade Glioma, identify the IDH status and the status of the 1p/19q codeletion, thus creating a classifier capable of predicting and differentiating 5 different molecular subtypes [26]. Additional noteworthy contributions to this field include the work of Pasquini et al., who used a 2D CNN on 6 different MRI modalities to discriminate between the IDH status of GBM patients with the best performance obtained on relative cerebral volume (rCBV) sequences (Accuracy = 83%), implying the underlining IDH pathophysiology [27]. Gore et al. created and tested a 3 pathway CNN on a TCIA dataset to discriminate the IDH status obtaining an accuracy of 94% [28]. Furthermore, Tupe-Waghmare et al. employed a semi-supervised hierarchical multi-task model trained on labeled and unlabeled data, having managed to simultaneously predict grade, IDH, 1p/19q and MGMT status with an average accuracy of 82% [29].

Although both Radiomics and Radiogenomics show great diagnostic and prognostic potential, there are still certain limitations. Most studies are retrospective, with no prospective validations (including this study) to confirm the performance of the algorithms and their ability to generalize. Additionally, Deep Learning models are generally trained using datasets from singular medical centers, however, in order to increase their performance, they need extensive, varied datasets [30]. Moreover, in most studies, the imaging used for the training of the algorithms has very high resolutions (3 T), while testing them on 1.5 T acquisitions shows greatly reduced performance. Our study obtained a comparable performance on 100% 1.5 T acquired images highlighting the generalizability of Choi’s model. As most of the studies on the topic, Choi’s algorithm uses 3 conventional MRI sequences which in essence are strictly anatomical. However, as Pasquini et al. have already shown in their study, perfusion modalities such as rCBV (due to their functional aspect) might prove even more suitable for IDH prediction [27].

The present study is also a proof-of-concept that applying very well documented pretrained AI models on foreign data is feasible and practicable even by non-specialists with little to no background in computer science. Our main pitfall on this was regarding a missing file from the repository. Thankfully, the author assisted us and the rest of the community who wanted to test or further develop their respective code.

To transform this idea into a clinical tool, a few more steps need to be taken. Perhaps a per center algorithm training would prove a more feasible start until a universally generalizable approach reaches comparable performance. Undoubtedly, the prediction code would need a user-friendly interface and constant feedback from the community of users. The data analyzed by each center would contribute to further improvement of the core algorithm.

A major limitation of our study is the small sample size. The markedly low number of patients recruited for this study stems from two factors. Firstly, there is a significant inhomogeneity regarding MRI protocols across departments and medical centers, and therefore 41 patients had incomplete cerebral imaging studies. Secondly, testing the presence of the IDH mutation is a rather newly introduced diagnostic test in our department and, due to the rarity of its detection, has not been employed for all patients. We however acknowledge this major drawback of our study and plan on overcoming these deficiencies in the upcoming continuation of this research, including performing IDH mutation testing for both future patients and the Formalin-Fixed Paraffin-Embedded (FFPE) glioblastoma (GBM) samples.

Recently, the Working Committee 3 of the Consortium to Inform Molecular and Practical Approaches to CNS Tumor Taxonomy (cIMPACT-NOW) has reconvened to establish a series of recommendations and considerations regarding the inclusion of several newly discovered CNS tumor types and subtypes in future classifications [31]. These proposed additions may help in future imaging-based prediction studies as our own, discriminating between the various subtypes and presence of specific genotypes, including pediatric-type glial/glioneuronal tumors. Nevertheless, despite the obvious benefits of utilizing such a complex and all-encompassing classification system in future studies, for the purpose of our current research it would have not made a significant change to the allotment of the tumors included.

Despite the evolution of this technology and the impact it can have on the medical field, the way Deep Learning algorithms work remains very abstract and may seem difficult to understand for those without the appropriate background. Facts such as these can diminish the trust of medical practitioners in these methods, making adoption in clinical practice more cumbersome. Similarly, many of the extracted radiomic features do not yet have a pathophysiological explanation [32]. A joint effort from both the medical and the IT communities would be required in order to develop reliable, trustworthy radiomic tools and high-quality studies.

## 5. Conclusions

The present external validation of the IDH prediction algorithm showed promising and comparable results to the TCIA external dataset from the original article. The presented method is easily accessible and affordable with opportunity for further development. However, radiogenomic methods are far from perfect due to technical and administrative issues that need to be addressed. External validation studies, such as the one presented, contribute to the evaluation of the original models, offer essential feedback, and, last but not least, promote and raise awareness regarding Radiomics and Radiogenomics within the scientific community and the research opportunities they may provide.

## Figures and Tables

**Figure 1 medicina-58-00526-f001:**
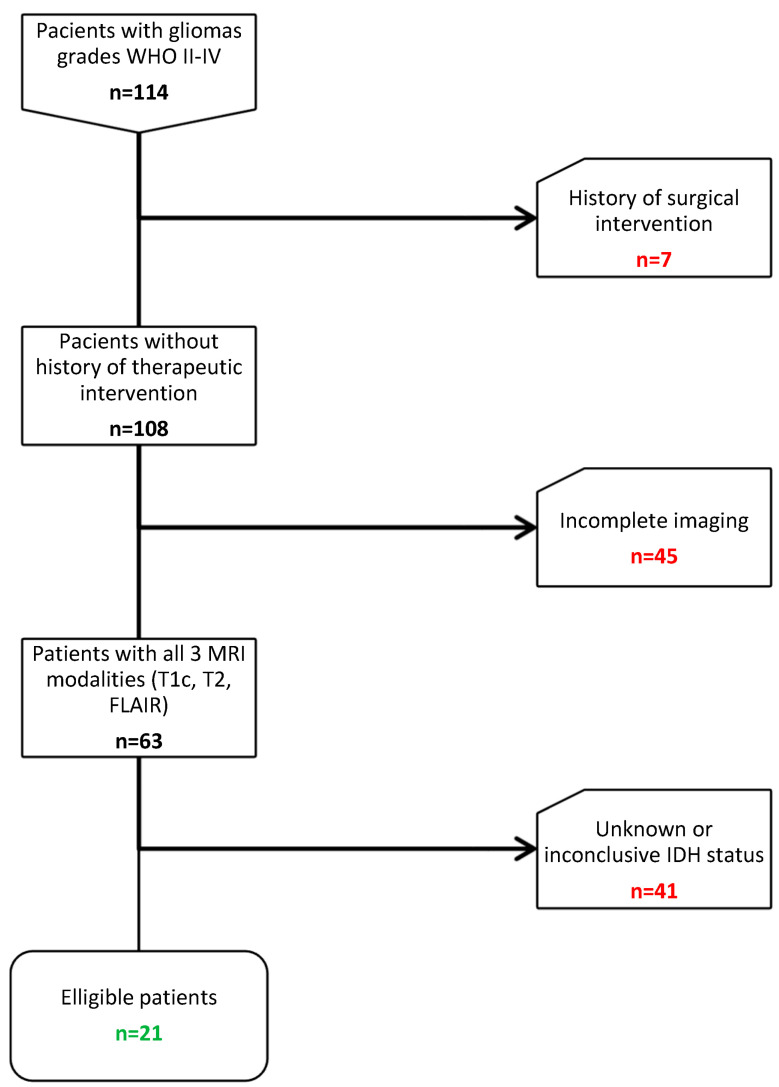
Sample selection diagram. Abbreviations: WHO, World Health Organization; MRI, magnetic resonance imaging; T1c, T1 contrast enhanced; FLAIR, fluid-attenuated inversion recovery; IDH, Isocitrate dehydrogenase.

**Figure 2 medicina-58-00526-f002:**
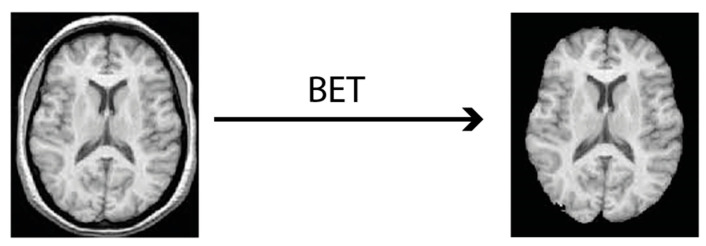
Skull stripping process using the Brain Extraction Tool (BET).

**Figure 3 medicina-58-00526-f003:**
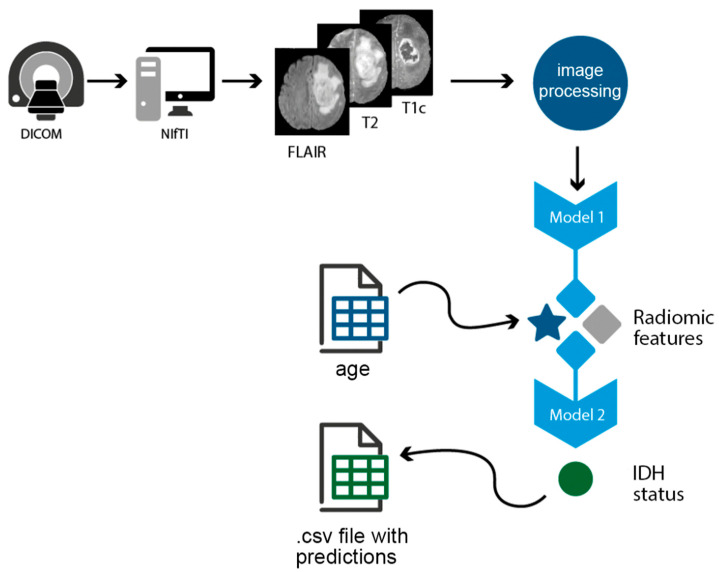
The pipeline of the python script that automated the testing. Abbreviations: DICOM, Digital Imaging and Communications in Medicine; NIfTI, Neuroimaging Informatics Technology Initiative; FLAIR, fluid-attenuated inversion recovery; T1c, T1 contrast enhanced; csv, Comma Separated Values; IDH, isocitrate dehydrogenase.

**Figure 4 medicina-58-00526-f004:**
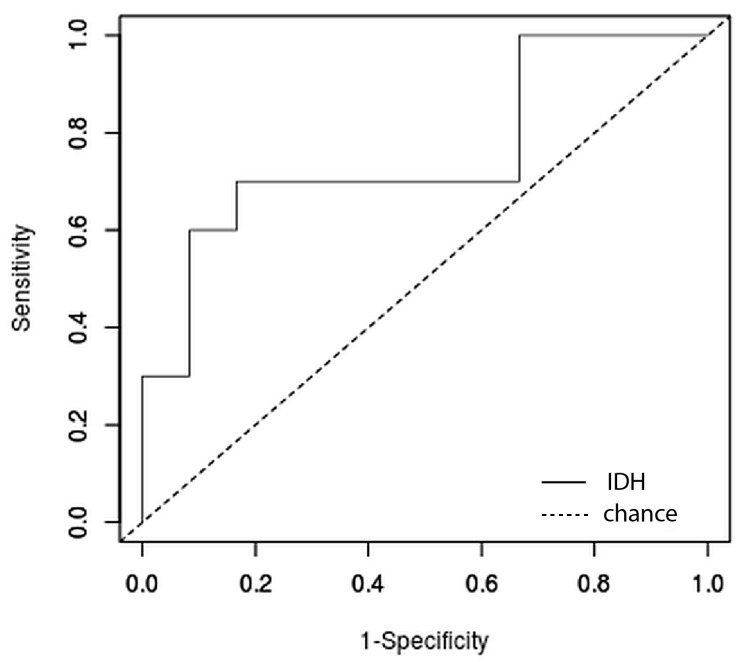
Receiver Operating Characteristic (ROC) curve of the IDH prediction test. IDH, Isocitrate Dehydrogenase.

**Table 1 medicina-58-00526-t001:** Comparison between our sample characteristics (descriptive statistics) and Severence training dataset.

Variable	CCECH (*n* = 21)	Severence (*n* = 856)	*p*-Value
Age (average)	48.6 ± 15.6	52.9 ± 15.5	0.2095 ^a^
Sex	12 f (57%)	492 f (57.5%)	0.0034 ^b^
	9 m (43%)	364 m (42.5%)	
IDH status	9 IDH+ (43%)	239 IDH+ (27.9%)	0.0008 ^b^
	12 IDH− (57%)	617 IDH− (72.1%)	
WHO grade	5 II (24%)	175 II (20.5%)	0.3123 ^b^
	3 III (14%)	169 III (19.7%)	
	13 IV (62%)	612 IV (59.8%)	

^a^ Unpaired student’s *t*-test; ^b^ Chi-square test. Abbreviations: CCECH, Cluj County Emergency Clinical Hospital; IDH, Isocitrate Dehydrogenase; WHO, World Health Organization.

**Table 2 medicina-58-00526-t002:** Contingency table between imunohistochemistry/molecular IDH status (Real) and its algorithm prediction (Predicted).

Real	IDH+	IDH−
Predicted		
IDH+	7	3
IDH−	2	9

Abbreviations: IDH, Isocitrate Dehydrogenase.

## Data Availability

Data may be obtained from the authors of this manuscript.

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
