# Peer review of "External Validation of a Convolutional Neural Network for IDH Mutation Prediction"

_medicina, 2022, doi:10.3390/medicina58040526_

Round 1
Reviewer 1 Report
This study is an external validation of an AI algorithm to predict IDH mutation status of glioma developed by Choi et al. The study is overall well-conducted and clinically relevant. A major limitation is its small sample size.
Major comments:
1. The sample size of this study is very small (only 21), yielding a large confidential interval (e.g., 0.53 to 0.91 for AUC). It would be more beneficial if the authors could expand their dataset to better evaluate the algorithm externally.
2. I recommend comparing the patient characteristics of this study with those of the training dataset (Severance Set) in Choi’s study. This can give us a better understanding of how this external validation dataset differs from the training set.
3. In the conclusions of the abstract, the authors state “performance can greatly be improved with a larger and more heterogeneous training set”. However, this statement is not supported by the data and is not a focus of this study. Instead, this can be a good argument in the discussion session.
4. I suggest adding more discussion about the authors’ experience of implementing an AI algorithm developed by another institution in another country, including all the challenges and pitfalls during the process, the feasibility and obstacles of using it in real clinical settings, and suggestions for the algorithm developers.
Minor comments:
Title: “IDH mutation prediction” would be more accurate.
Abstract: The background part is too long and can be shortened.
Line 34-35: Please include the essential patient characteristics in the results and report the 95% confidential intervals for the statistics.
Line 68-69: “reduces sampling errors” would be more prudent. Also, biopsy is still needed as pathology will still be the gold standard in the foreseeable future.
Line 123-125: Please provide the manufacturer and slice thickness of the images.
Figure 2: In my opinion, this figure is unnecessary as the content is straightforward and has been described well in the text. The authors may choose to move this figure to supplementary material.
Line 156-158: “… together with T1C and T2 images and patient age”
Table 1: The data on sex, IDH status, and WHO grade all indicates there are 22 patients in total, differing from other parts of the article.
Line 183: Please report the 95% confidential interval for accuracy.
Line 199: The SNUH and TCIA sets are both for external validation in Choi’s study. One difference could be TCIA is from another country.
Line 204-207: Image protocol and patient characteristics can be other sources of difference.
Line 224: The abbreviation “GBM” hasn’t been defined.
Line 221, 226: Are these data from internal or external validation?
Line 247-250: This argument is overstated. The authors should make the conclusions stick to their study and interpret the results with prudence. It would be more appropriate to say “the presented method is easily accessible and affordable with opportunity for further development.”
Author Response
We are very thankful to the reviewers for their thorough analysis. In light of their useful comments, we have revised the present research paper in the hope that this version will be considered satisfactory. We have addressed all the comments as explained below.
Reviewer #1:
This study is an external validation of an AI algorithm to predict IDH mutation status of glioma developed by Choi et al. The study is overall well-conducted and clinically relevant. A major limitation is its small sample size.
Major comments:
- The sample size of this study is very small (only 21), yielding a large confidential interval (e.g., 0.53 to 0.91 for AUC). It would be more beneficial if the authors could expand their dataset to better evaluate the algorithm externally.
Response: Thank you for your valuable comments. We agree that this is the major limitation of our proof-of-concept study. However, as we have now added an explanation in the discussion, there was a significant heterogeneity in the protocols of brain MRI across the departments and medical centers that provided the imaging studies for our patients. Furthermore, testing the IDH mutation has only been introduced in our department within the last few years and has not been employed for all cases. We do however intend to perform it more frequently, including in the FFPE specimens that posses adequate imaging studies, for the upcoming continuation of our research.
- I recommend comparing the patient characteristics of this study with those of the training dataset (Severance Set) in Choi’s study. This can give us a better understanding of how this external validation dataset differs from the training set.
Response: We have added the requested comparison in the table.
- In the conclusions of the abstract, the authors state “performance can greatly be improved with a larger and more heterogeneous training set”. However, this statement is not supported by the data and is not a focus of this study. Instead, this can be a good argument in the discussion session.
Response: We have moved this point to the discussion instead of the conclusions.
- I suggest adding more discussion about the authors’ experience of implementing an AI algorithm developed by another institution in another country, including all the challenges and pitfalls during the process, the feasibility and obstacles of using it in real clinical settings, and suggestions for the algorithm developers.
Response: We have added two additional paragraphs on this aspect in the discussion.
Minor comments:
Title: “IDH mutation prediction” would be more accurate.
Response: We changed the title to make it more straightforward, with the inclusion of your suggestion.
Abstract: The background part is too long and can be shortened.
Response: We have shortened the background part. We hope you will find it suitable.
Line 34-35: Please include the essential patient characteristics in the results and report the 95% confidential intervals for the statistics.
Response: We have added the required patient characteristics.
Line 68-69: “reduces sampling errors” would be more prudent. Also, biopsy is still needed as pathology will still be the gold standard in the foreseeable future.
Response: We made the appropriate adjustment to the text.
Line 123-125: Please provide the manufacturer and slice thickness of the images.
Response: We have provided the manufacturers (incl. percentages of patients respective of each apparatus), as well as the corresponding thickness of MRI slices for the investigations performed.
Figure 2: In my opinion, this figure is unnecessary as the content is straightforward and has been described well in the text. The authors may choose to move this figure to supplementary material.
Response: We have removed the figure entirely, adding it as a graphical abstract if deemed suitable.
Line 156-158: “… together with T1C and T2 images and patient age”
Response: We have made the change.
Table 1: The data on sex, IDH status, and WHO grade all indicates there are 22 patients in total, differing from other parts of the article.
Response: Thank you for pointing this out, we have since made the correction. We have also added the tests used for p-values.
Line 183: Please report the 95% confidential interval for accuracy.
Response: The 95% CI is now added.
Line 199: The SNUH and TCIA sets are both for external validation in Choi’s study. One difference could be TCIA is from another country.
Response: We modified the sentence into: We compared our results with those obtained from the abovementioned datasets (TCIA, SNUH), as our validation is also an external one.
Line 204-207: Image protocol and patient characteristics can be other sources of difference.
Response: We have added the statement in the text.
Line 224: The abbreviation “GBM” hasn’t been defined.
Response: We have added the definition of this abbreviation.
Line 221, 226: Are these data from internal or external validation?
Response: We have now mentioned that the data was from an internal validation.
Line 247-250: This argument is overstated. The authors should make the conclusions stick to their study and interpret the results with prudence. It would be more appropriate to say “the presented method is easily accessible and affordable with opportunity for further development.”
Response: We completely agree with your remark and have modified the sentence in the conclusions.
Again, we would like to thank the reviewers for their valuable input and suggestions and hope that the modifications we made are appropriate.

Reviewer 2 Report
The study presents an external validation of a deep-learning algorithm for IDH prediction in gliomas. Although the paper in consideration (Choi et al.) already described an external validation with much more cases than the present paper, this study offers an interesting point for prediction on small number of patients and on 1.5T scanner, which is not frequent in these studies. I have some comments:
The title does not go straight to the point. Being an external validation of a previous study the readers should be immediately aware of the purposes of the study. Something like "External validation of Convolutional Neural Network for IDH prediction" sounds more straightforward.
Both the introduction and discussion would benefit of an updated literature. In particular, in addition to WHO 2016 classification, it would be useful cite the cIMPACT recommendations (10.1111/bpa.12832) which gave much more relevance to glioma molecular profile. With this regards, how do the tumors in this study are classified (please change table 1 accordingly)?
Some new IDH prediction models have been recently described and that are worth citing (10.1109/ACCESS.2021.3136293; 10.3906/elk-2104-180; 10.3390/jpm11040290). In particular, Pasquini et al. used a 1.5T scanner, an issue that has been addressed in discussion by the authors as well.
Choi et al. focused only on conventional sequences as many other studies, while the use of advanced sequences could be useful in these predictive model. Could the authors please discuss this point?
Being the Choi study already been externally validated, the present paper discussion should highlight more the novelty of testing the present algorithm on 1.5T obtaining similar results, thus resulting in more generalizability of this model.
Minor:
It would be better if the abbreviations would be in brackets at first rather than the noun.
Please improve quality of Figure 5.
Author Response
We are very thankful to the reviewers for their thorough analysis. In light of their useful comments, we have revised the present research paper in the hope that this version will be considered satisfactory. We have addressed all the comments as explained below.
Reviewer #2:
The study presents an external validation of a deep-learning algorithm for IDH prediction in gliomas. Although the paper in consideration (Choi et al.) already described an external validation with much more cases than the present paper, this study offers an interesting point for prediction on small number of patients and on 1.5T scanner, which is not frequent in these studies. I have some comments:
The title does not go straight to the point. Being an external validation of a previous study the readers should be immediately aware of the purposes of the study. Something like "External validation of Convolutional Neural Network for IDH prediction" sounds more straightforward.
Response: Thank you for this remark, we have modified the title in a manner that we hope is more to the point.
Both the introduction and discussion would benefit of an updated literature. In particular, in addition to WHO 2016 classification, it would be useful cite the cIMPACT recommendations (10.1111/bpa.12832) which gave much more relevance to glioma molecular profile. With this regards, how do the tumors in this study are classified (please change table 1 accordingly)?
Response: We have updated the literature and have included the article specified in the references. We also touched on the benefits of using the suggested cIMPACT classification system for future studies within a new paragraph in the discussions. However, after having considered the modifications proposed in this article, we feel that it would change neither the classification of the tumors included in our study (being only adults having the IDH mutation tested), nor the details presented in table 1. We however hope that our justification was strong enough.
Some new IDH prediction models have been recently described and that are worth citing (10.1109/ACCESS.2021.3136293; 10.3906/elk-2104-180; 10.3390/jpm11040290). In particular, Pasquini et al. used a 1.5T scanner, an issue that has been addressed in discussion by the authors as well.
Response: We have added the recommended citations within the discussions.
Choi et al. focused only on conventional sequences as many other studies, while the use of advanced sequences could be useful in these predictive model. Could the authors please discuss this point?
Response: We have discussed this point, arguing that “perfusion modalities such as rCBV (due to their functional aspect) might prove even more suitable for IDH prediction,” as shown by Pasquini et al.
Being the Choi study already been externally validated, the present paper discussion should highlight more the novelty of testing the present algorithm on 1.5T obtaining similar results, thus resulting in more generalizability of this model.
Response: We have highlighted that this is a proof of concept study and that it shows the generalizability of this model, proposing even some suggestions to transform this into a clinical tool.
Minor:
It would be better if the abbreviations would be in brackets at first rather than the noun.
Response: We have made this modification where applicable.
Please improve quality of Figure 5.
Response: As we have removed figure 2, this now became figure 4 and we have improved its quality.
Again, we would like to thank the reviewers for their valuable input and suggestions and hope that the modifications we made are appropriate.
Round 2
Reviewer 2 Report
The scientific content of the paper improved significantly after the revisions.
Here I address some minor issues:
- The two new paragraphs added in the discussion would fit more in another part of the paper. Both would be better in the last part of the discussion, as they address limitation of the study and future directions.
- Vendors of the MR scanners have been added but it would be better to write the model of the scanner and the vendor in brackets as it is usual custom.
- Please provide explanation for all abbreviation in figures. Figures should be readable even without reading the main text.
- Please revise the english for improving readability
Author Response
We are very thankful to the reviewers for their thorough analysis. In light of their useful comments, we have revised the present research paper in the hope that this version will be considered satisfactory. We have addressed all the comments as explained below.
Reviewer #2:
The scientific content of the paper improved significantly after the revisions.
Here I address some minor issues:
- The two new paragraphs added in the discussion would fit more in another part of the paper. Both would be better in the last part of the discussion, as they address limitation of the study and future directions.
Response: Once again, thank you kindly for your remarks and suggestions. We have moved the two paragraphs at the end of the discussions and have arranged the references accordingly.
- Vendors of the MR scanners have been added but it would be better to write the model of the scanner and the vendor in brackets as it is usual custom.
Response: We have now included a supplementary file with the respective model of the scanner and sequence characteristics for each of the patients. We hope this is acceptable.
- Please provide explanation for all abbreviation in figures. Figures should be readable even without reading the main text.
Response: We have added the explanations for all abbreviations within the figures and text.
- Please revise the english for improving readability
Response: We made a broad revision of the English language in the text. We hope you will find it improved.
